# Transcriptomics Integrated with Changes in Cell Wall Material of Chestnut (*Castanea mollissima* Blume) during Storage Provides a New Insight into the “Calcification” Process

**DOI:** 10.3390/foods11081136

**Published:** 2022-04-14

**Authors:** Yu Chen, Cancan Zhu, Yuqiang Zhao, Shijie Zhang, Wu Wang

**Affiliations:** Institute of Botany, Jiangsu Province and Chinese Academy of Sciences, Nanjing 210014, China; 15150530195@163.com (Y.C.); zhaoyuqiang123@126.com (Y.Z.); zsjtd310@126.com (S.Z.); 2017204015@njau.edu.cn (W.W.)

**Keywords:** chestnut, calcification, cellulose, lignin, pectin, transcriptome

## Abstract

Chestnut “calcification” is the result of a series of physiological and biochemical changes during postharvest storage; however, the associated mechanisms are unclear. In this study, several potential calcification-related physicochemical parameters in chestnut, including moisture, cell wall materials, cellulose, lignin, and pectin, were measured. Transcriptome analysis was performed on chestnut seeds during different stages of storage. The results showed that the degree of calcification in the chestnut seeds was significantly negatively correlated with the moisture content (r = −0.961) at room temperature (20–25 °C) and a relative humidity of 50–60%. The accumulation of cell wall material in completely calcified seeds was 5.3 times higher than that of fresh seeds. The total content of cellulose and lignin increased during the storage process. Transcriptome analysis of 0% and 50% calcified chestnut was performed; a total of 1801 differentially expressed genes consisting of 805 up-regulated and 996 down-regulated genes were identified during the calcification process. Furthermore, response to water, water deprivation, and salt stress were most enriched by gene ontology (GO) and gene set enrichment analysis (GSEA). The Kyoto Encyclopedia of Genes and Genomes (KEGG) pathways related to chestnut calcification included purine metabolism, RNA degradation, the mRNA surveillance pathway, starch and sucrose metabolism, arginine and proline metabolism, and fatty acid metabolism, and were detected. Most of the genes involved in cellulose synthase, lignin catabolism, and pectin catabolism were down-regulated, while only two important genes, scaffold11300 and scaffold0412, were up-regulated, which were annotated as cellulose and pectin synthase genes, respectively. These two genes may contribute to the increase of total cell wall material accumulation during chestnut calcification. The results provided new insights into chestnut calcification process and laid a foundation for further chestnut preservation.

## 1. Introduction

Chestnut is a species of beech (*Castanea*, *Fagaceae*) and is one of the most economically important tree species in China, where it is used as a woody grain and a hardwood crop [1]. Castanea is a relatively small genus, with only about 13 species worldwide; its members are widely distributed in the subtropical zone of the northern hemisphere [2]. Chestnuts include *Castanea mollissima* Blume, known as Chinese chestnut, *Castanea seguinii* Dode, and *Castanea henryi* (Skan) Rehder and E. H. Wilson, among which only Chinese chestnut is widely cultivated. Chestnut fruit is rich in starch, fat, protein, and multivitamins. It also contains carotene as well as riboflavin, niacin, and other vitamins [3]. Chestnut seeds (also called chestnuts) can be eaten fresh and fried or can be made into a variety of delicacies [4]. However, the seed is recalcitrant, and thus it is prone to moisture loss and deterioration during storage, leading to calcification and a loss of commercial value.

Calcification is a physiological disorder of the chestnut caused by the storage environment. It is a deterioration accompanied by white and hard of chestnut pulp, which could result in loss edible value [5]. Many studies addressed the physiological and biochemical aspects of chestnut calcification. Gu et al. [6] reported that the occurrence of chestnut calcification was related to changes in the crystalline structure of starch and its thermal and textural characteristics, which were caused by the degradation of starch as a result of increased amylase activities induced by moisture loss; however, no correlation was observed between the sucrose content and chestnut calcification. Wen et al. [5] investigated the relationship between the moisture loss rate and plasma membrane lipid oxidation in chestnut calcification. At low humidity, the activities of plasma membrane lipase and lipoxygenase in chestnut were higher than those at high humidity, which implied that the plasma membrane lipid oxidation induced by rapid moisture loss might be closely related to the occurrence of calcification.

The maturation and senescence of fruit are associated with the relative composition of the cell wall materials, which plays an important role in determining the flavor of fruit [7]. Calcification causes the taste of chestnut to gradually deteriorate and finally lose its edible value. Recent studies confirmed that chestnut calcification might be mainly induced by the loss of moisture and related to a disruption of the cell wall materials [5,8]. The main components of plant cell walls include pectin, cellulose, hemicellulose, and lignin [9]. Cellulose in the primary wall imparts rigidity and tension to the edible and non-edible tissues [10]. Pectin polysaccharides in the cell wall include galacturonic acid, rhamnose, arabinose, and galactose, which confer plasticity and allow the expansion of the fruit. Lignin affects the rigidity and cohesion of edible seeds [11].

Few studies have explored chestnut calcification especially at the molecular level. The variations in the metabolic network including a significant decrease in some monosaccharides and increase in unsaturated fatty acids, amino acids, lignin precursors and antioxidants were determined by widely targeted metabolomic analysis of chestnut calcification [12]. In this study, the relationship between the rate of moisture loss in chestnut and the pectin, cellulose, and lignin contents in the cell wall was investigated to explore the physiological and biochemical aspects of calcification. The transcriptome of chestnut subjected to different calcification treatments were analyzed to better understand the changes in transcript expression during calcification.

## 2. Materials and Methods

### 2.1. Plant Materials and Treatments

The tested chestnut variety, *C.*
*mollissima* ‘Hongli’ [13], was planted as part of the Chinese chestnut germplasm resources research area, Nanjing, Jiangsu Province, China. The seeds were collected, freshly cracked, and shelled. The seeds were transported to the laboratory where diseased seeds were removed. Ten kilograms of uniform seeds in same size were placed in one layer and stored at room temperature (20–25 °C) at a relative humidity of 50–60%. Thirty seeds were randomly sampled every 5 days (d), and after careful removal of the outer shell, the seeds were cut longitudinally; next, the calcification rate was calculated according to the ratio of the area of calcification to the area of the incision.

The degree of calcification of chestnut was partitioned into five grades based on the calcified area, including 0%, 1–25%, 26–50%, 51–75%, and 76–100% of the incision area, which defined the stages of calcification in subsequent experiments. The degree of calcification was calculated according to Equation (1) as described by Zhang et al. [14].
(1)$$\mathrm{Calcification}\;\left(\%\right)=\frac{\Sigma \;\left(\mathrm{calcification}\;\mathrm{grade}\;\times \;\mathrm{seed}\;\mathrm{number}\right)}{5\;\times \;\mathrm{seed}\;\mathrm{number}}\times 100\;$$

### 2.2. Determination of the Moisture Content

Seed moisture content was measured by drying sample [15]. The chestnut materials from each sampling stages were placed in a bottle and weighed. The temperature of the drying oven was preset to 105 °C, the fresh samples were placed in a dry box and dried to a constant weight. Then, the bottle was taken out and the dry seeds was weighed. Each sample was measured three times. The moisture contents were expressed as a percentage of water in fresh and dry seeds according to Equation (2).
(2)$$\;\mathrm{Moisture}\;\left(\%\right)=\frac{\mathrm{fresh}\;\mathrm{weight}-\mathrm{dry}\;\mathrm{weight}}{\mathrm{fresh}\;\mathrm{weight}}\times 100$$

### 2.3. Determination of Cell Wall Material in Different Calcified Chestnut Seeds

#### 2.3.1. Extraction of the Cell Wall Material

The material of the chestnut cell wall was extracted [16]; 0.1 g of 0%, 25%, 50%, 75%, and 100% calcified chestnut seeds were obtained. After grinding and filtering, 80% (*v*/*v*) ethanol was added to the samples, which were then heated until dissolved, and 90% (*v*/*v*) dimethyl sulfoxide was added and filtered. The filtrate was used as the cell wall material after a vacuum drying process. The experiment was repeated three times. The results were expressed as g kg^−1^ based on the fresh weight.

#### 2.3.2. Extraction and Determination of the Lignin and Cellulose Contents

Lignin was measured as Klason lignin according to the method presented by Femenia et al. [17]. A total of 0.1 g powder sample was homogenized in 20 mL 72% (*v*/*v*) H_2_SO_4_ solution for 4 h. Then, the mixed solution was transferred to a 1000 mL flask and added 765 mL distilled water. The flask was heated in boiling water for 2 h. Then, the mixed sample was filtered with a crucible of known weight (W1) and was dried in an oven until the weight remained constant (W2). The content of lignin was calculated according to the formula: lignin (%) = (W2 − W1)/0.1 × 100%. The content of cellulose was measured using the method presented by Huang et al. [18]. First, 0.1 g powder of sample was continuously shaken in a 50 mM Tris: HCl solution with 1% SDS over 3 h at 20 °C. Then, the mixed solution was centrifuged at 20,000× *g* for 15 min, washing with water, acetone and air-drying, the residue was taken to incubate in 5 mL 2 M trifluoroacetic acid at 120 °C for 90 min. After incubating, the residue was washed with water and ethanol and was dissolved with 60% (*v*/*v*) H_2_SO_4_ at 4 °C for 30 min. Finally, the cellulose content was determined colorimetrically after being diluted appropriately using anthrone as a coloring agent. The concentration of cellulose content was determined by standard curves measured in glucose. The concentrations of lignin and cellulose were expressed as g kg^−1^ based on the fresh weight.

#### 2.3.3. Extraction and Determination of Pectin Content with Different Solubilities

Water-soluble pectin (WSP), chelate-soluble pectin (CSP), and ionically soluble pectin (ISP) were extracted using the improved method presented by Kintner et al. [19]. Two milliliters of different calcified chestnut seed extraction solutions were obtained and combined with 8 × 10^−3^ L concentrated sulfuric acid in a water bath for 30 min; then, 0.5 × 10^−3^ L ethanol solution was added to the mixture from the last step and incubated for 30 min at room temperature in darkness. At the end of the incubation period, absorbance was measured using a UV-Vis spectrophotometer (Thermo Scientific™ Evolution™ 350, Waltham, MA, USA) at a wavelength of 530 nm. The results were expressed as g kg^−1^ based on the fresh weight.

### 2.4. Transcriptome Analysis of Chestnut with Different Degrees of Calcification

#### 2.4.1. RNA Extraction and Transcriptome Sequencing

The RNA was extracted from 0%, 50%, and 100% calcified chestnut fruit using the Column Plant RNA Out kit (Fuji, Chengdu, Sichuan, China). However, RNA could not be obtained from the seeds, which were 100% calcified because they had been seriously damaged; thus, only 0% and 50% calcified seeds were used for library construction and sequencing. Sequencing was performed with an Illumina (San Diego, CA, USA) HiSeq^TM^ 2500 platform according to standard procedures. The low quality or poly-N tailed reads were removed to obtain clean reads which were mapped to the chestnut reference genome (https://www.hardwoodgenomics.org/Genome-assembly/1962958, accessed on 17 April 2019). The mapping and assembly used Tophat and Cuffling software, respectively [20]. The fragments per kb per million reads (FPKM) values were calculated to estimate the expression profile of each gene with RSEM tools.

#### 2.4.2. Analysis of Differentially Expressed Genes

To identify the differentially expressed genes (DEGs) of the 0% and 50% calcified seeds, the edgeR package (http://www.r-project.org/, accessed on 17 April 2019) was used. Genes with a fold-change ≥2 and a false discovery rate (FDR) <0.05 were identified as differentially expressed between two samples. All of the mapped unigenes were compared with public databases (Non-redundant (Nr), Swiss-Prot, Clusters of Orthologous Genes (COG), and Pfam) using BLASTX (http://blast.ncbi.nlm.nih.gov/Blast.cgi accessed on 6 Jun 2019) with an E-value of 1×10^−5^ [21]. The DEGs were mapped to gene ontology (GO) terms in the GO database (http://www.geneontology.org/, accessed on 6 June 2019); in addition, significantly enriched GO terms in the DEGs compared to the genomic background were defined by a hypergeometric test. The GO terms were defined as significantly enriched when *p* ≤ 0.05. The DEGs were mapped to the Kyoto Encyclopedia of Genes and Genomes (KEGG) database for identifying significant enriched metabolic pathways or signal transduction pathways. KEGG pathways with *p* ≤ 0.05 were identified as significantly enriched.

#### 2.4.3. Gene Set Enrichment Analysis

To explore the potentially significant biological pathways related to calcification of chestnut fruit, gene set enrichment analysis (GSEA, v2–2.2.3) was performed using mRNA-seq data from the reference gene set (GO gene set). The normalized enrichment score was calculated, and the number of permutations was set at 1000. The results were significant if they met the criteria of a nominal *p*-value < 0.05 and FDR < 0.25 [22].

#### 2.4.4. Quantitative Real-Time PCR (qRT-PCR) Analysis

Fifteen DEGs were randomly selected to validate the transcriptome sequencing by qRT-PCR. Following RNA extraction, first-strand cDNA was synthesized using a Prime Script RT Reagent Kit (Takara, Kusatsu, Japan). The primers were designed by primer6 software, and CMACT was used as an internal control. The 20 µL qRT-PCR reactions included 10 µL SYBR Premix Ex Taq (2×), µL cDNA, 2 µL primers (F/R), and 6 µL distilled water. The PCR program was as follows: amplification curve of 95 °C for 30 s, followed by 40 cycles at 95 °C for 10 s, 60 °C for 30 s, and 72 °C for 30 s; with a melting curve at 95 °C for 15 s, 60 °C for 60 s, and 95 °C for 15 s. The relative expression level of each gene was calculated using the method of 2^−^^∆∆Ct^ [23].

#### 2.4.5. Statistical Analyses

Data analyses were performed in SPSS version 19.0 (IBM Corp, Armonk, NY, USA) and Excel 2013 software (Microsoft Corp, Redmond, WA, USA), and all data were expressed as the mean ± standard deviation. One way analysis of variance (ANOVA) was employed to determine the statistical difference. Significant differences between means were identified using Duncan’s multiple range test (*p* < 0.05). All experiments were repeated at three times.

## 3. Results

### 3.1. Relationship between the Degree of Chestnut Calcification and Moisture Content

After 20 d at room temperature, the moisture content of the chestnut seeds decreased from 45% to 19%, a reduction of 59%. The relatively high rate of water loss appeared at 5–15 d of storage. On 30 d, the degree of calcification was 79%, when the moisture content was reduced to 14% (Figure 1). Statistical analysis showed that the degree of calcification was negatively correlated with the moisture content (*r* = −0.961, *p* < 0.01). On 47 d, the chestnut was completely calcified, with only 11% the moisture content.

### 3.2. Changes in Cell Wall Material (CWM) Content in Different Stages of Chestnut Calcification

The CWM content of chestnut in different stages of calcification (Figure 2A) varied significantly (aside from those 25% calcified) (Figure 2B). With an increase in the degree of chestnut calcification, the fruit surface became yellow brown, and the interior became hard and white. The fresh chestnut seeds had the lowest CWM content of 96.2 ± 21.0 g kg^−1^, while the CWM of the 100% calcified seeds was 506.9 ± 17.7 g kg^−1^, which was 5.3 times higher than that of the fresh chestnut seeds. The results indicated that the chestnut calcification process involved a reduction in the moisture content and a constant accumulation of CWM in the chestnut fruit.

### 3.3. Changes in the Lignin and Cellulose Content of Different Stages of Calcification

The changes in lignin and cellulose in increments of 0.1 g of chestnut fruit were determined; the results are shown in Figure 2C. With an increase in the seed calcification rate, the lignin content of the chestnut fruit increased at early stages and then sharply decreased. When the calcification rate was 25%, the lignin content reached the highest value of 46.9 ± 8.7 g kg^−1^, compared with 31.6 ± 4.5 g kg^−1^ in the fresh fruit. However, the lignin content decreased sharply to 18.3 ± 2.4 g kg^−1^ at 50% calcification and then started to increase, while at 100% calcification, the lignin content increased to 35.8 ± 8.3 g kg^−1^. The change trend of the cellulose content in chestnut was opposite to the change in lignin content (r = − 0.856, *p* < 0.01), showing a decreasing tendency after an initial increase. The cellulose content of the fresh fruit was 25.3 ± 8.6 g kg^−1^, which decreased to 22.1 ± 7.4 g kg^−1^ mg at 25% calcification, but then rapidly increased to 52.6 ± 11.2 g kg^−1^ at 50% calcification. These values showed a downward trend under 75% and 100% calcification, finally decreasing to 39.4 ± 7.1 g kg^−1^. In general, in our study the total cellulose and lignin contents showed an opposite growth trend during the storage process.

### 3.4. Changes in the Pectin Content of Chestnut at Different Stages of Calcification

Pectic polysaccharides are the main components of the cell wall and confer plasticity and stretchability characteristics to the fruit. The pectin contents, including WSP, CSP, and ISP, were measured. The WSP had the highest pectin content, followed by CSP and then ISP (Figure 2D). During the process of chestnut calcification, the WSP content increased slowly compared with the initial time of storage, and the CSP decreased slightly while the IPS content increased considerably. Compared with the fresh chestnut fruit, the WSP content of the 100% calcified seed increased from 93.2 ± 8.5 g kg^−1^ to 125.9 ± 5.1 g kg^−1^, an increase of 35%. The content of CSP initially showed a sharp decrease during the process of calcification and then slowly increased. When the calcification ratio of the seeds was 50%, the content of CSP decreased to its lowest point (56.5 ± 5.4 g kg^−1^), accounting for 58% of the fresh seeds. With an increase in the calcification ratio, the ISP content decreased sharply to 16.7 ± 7.7 g kg^−1^, which accounted for 26% of the fresh seed. The WSP, CSP, and ISP all decreased at 50% calcification and increased beyond 50% calcification. This indicate that 50% calcification may be the key time point when the physiological metabolism of chestnut begins to appear disordered.

### 3.5. Transcriptome Sequencing and Assembly

Six samples named T1a, T1b, and T1c (0% calcification) and T3a, T3b, and T3c (50% calcification) were prepared for sequencing. Low-quality bases and non-carrier errors were removed from the raw data, resulting in 32,619,737, 30,425,950, 23,793,560, 31,853,069, 34,018,941, and 32,735,082 clean reads, respectively. The clean reads had Q30 quality scores higher than 93.1% (Table 1). After removal of the low-quality reads, the clean reads were mapped to the Chinese chestnut reference genome (https://www.hardwoodgenomics.org/Genome-assembly/1962958, accessed on 17 April 2019). More than 83.6% of the reads were mapped to the genome and 53,664,614, 49,885,627, 39,123,856, 53,975,213, 57,572,054, and 56,031,375 reads were uniquely mapped to the reference genome (Table 2).

### 3.6. Comparisons between the 0% and 50% Stages of Calcification

Transcriptional response to chestnut calcification were determined by comparing the transcriptomes of the 0% (T1) and 50% (T3) groups. All gene expression profiles in the two groups are shown as a heatmap (Figure 3 and Appendix A). We compared the variations in gene expression between 0% and 50% calcification; a total of 1801 DEGs, including 805 up-regulated and 996 down-regulated genes were identified (Appendix A). All DEGs were identified using a cutoff of |log_2_FC| ≥ 1 and FDR < 0.05.

### 3.7. GO and KEGG Enrichment of the DEGs

A total of 7864 annotated DEGs were classified into three GO categories: biological process, cellular component, and molecular function (Figure 4 and Appendix A). The largest GO terms in the biological process were the metabolic process, cellular process, and single-organism process. The top two GO terms in the molecular function were catalytic activity and binding. In the cellular component category, the top four GO terms were cell part, cell, organelle, and membrane part.

The GO enrichment analysis of the DEGs showed eight enriched terms, such as response to organonitrogen compounds, response to water, response to water deprivation, and carbohydrate catabolic process (Figure 5). The genetic responses to water and water deprivation at 0% and 50% calcification varied.

In this study, KEGG enrichment analysis of DEGs indicated the existence of 58 enriched KEGG pathways (Figure 6 and Appendix A). The most enriched pathways included purine metabolism, RNA degradation, the mRNA surveillance pathway, starch and sucrose metabolism, arginine and proline metabolism, glutathione metabolism, and fatty acid metabolism. Ten of the 13 DEGs under the purine metabolism pathway were up-regulated at 50% calcification. Moreover, a large part of the DEGs in these enriched pathways were up-regulated, such as those involved in RNA degradation (7/9), mRNA surveillance (5/6), arginine and proline metabolism (5/6), and glutathione metabolism (8/9). However, six out of eight of the DEGs annotated as starch and sucrose metabolism were down-regulated at 50% calcification.

### 3.8. GSEA Enrichment Analysis

The GSEA enrichment analysis was performed on all differentially expressed genes (0% and 50% calcification) based on the 64 enriched GO terms (Appendix A). As shown in Figure 7 and Figure 8, a total of 14 GO annotations were enriched during chestnut calcification, i.e., plant organ morphogenesis, cell wall organization or biogenesis, regulation of biological quality, nucleobase-containing small molecule metabolic process, seed development, response to osmotic stress, and response to salt stress (all *p* < 0.05). Responses to osmotic and salt stresses were enriched and were closely related to the water loss of chestnut during the calcification process.

### 3.9. Transcriptomic Insights into Chestnut Calcification and Moisture Content

Chestnut calcification is a process of continuous water loss. In the transcriptome data, water-related pathways (e.g., response to water, response to water deprivation, response to salt stress) were most enriched by GO and GSEA enrichment analysis. The moisture content of chestnuts decreased from 46.32% to 28.65% when compared with the fresh and 50% calcification seeds. The DEGs related to the moisture content in chestnut were observed in water transport (GO:0006833) and response to water deprivation (GO:0009414) pathway. Five DEGs (scaffold00315, scaffold02192, scaffold06454, scaffold05303, and scaffold10774) of water transport pathway, which were negatively correlated with moisture content, were up-regulated, except scaffold00555 was down-regulated (Figure 9A). In response to water deprivation pathway, the six DEGs (scaffold02492, scaffold00766, scaffold00962, scaffold03362, scaffold02192, and scaffold01872) were up-regulated and negatively correlated with water content (Figure 9B). The current data indicated that the expression of genes related to water transport and water deprivation were most active in 50% calcification, which might explain the reason for water loss during chestnut calcification.

### 3.10. Transcriptomic Insights into Cellulose and Lignin Metabolism

The base structure of cellulose is a polysaccharide composed of D-glucose with a *β*-1, 4-glycosidic bond. Cellulose synthase is located on the cell membrane and is organized by specific plasma membrane-bound cellulose synthase (CESA; EC 2.4.1.12) complexes in symmetrical rosette forms [9]. In this study, we detected six DEGs related to CESA in the cellulose biosynthetic process (GO: 0030244). Among them, scaffold01281, scaffold01202, scaffold02563, and scaffold11300 showed relatively higher expression levels (Table 3). The other two, named scaffold00980 and scaffold03287, displayed relatively lower expression levels. The expression levels of scaffold01281, scaffold01202, scaffold03287, and scaffold02563 at 0% calcification were all higher than those at 50% calcification, and only scaffold11300 maintained a higher expression level at 50% calcification. Six genes participated in cellulose synthesis; only scaffold11300 was up-regulated in the 50% calcified samples, which suggests that scaffold11300 is an important gene involved in cellulose anabolism. Sucrose synthase (SUS; EC 2.4.1.13) is another important enzyme involved in cellulose synthesis that provides UDP-glucose for cell wall cellulose synthesis [24]. In our study, we also found a SUS-related gene, scaffold0125, which showed higher expression levels in the 50% calcified samples. Sucrose synthase is reported to be related to the CESA complex and contributes to cellulose biosynthesis [16], which may contribute to cellulose synthesis and promote the accumulation of cellulose in the cell wall. Lignin is a complex phenylpropane monomer polymer that affects the rigidity and cohesion of the cell walls [25]. Here, we detected two genes, scaffold34956 and scaffold00325, in the lignin catabolic process (GO: 0046274). Scaffold34956 showed a slightly higher expression level, while scaffold00325 displayed quite a low expression level. The two lignin-related genes were down-regulated at 50% calcification; from the aforementioned results, we proposed that the lignin was catabolically inactive, which could also explain why the physiological data displayed the lowest lignin content at 50% calcification.

### 3.11. Pectin Metabolism-Related DEGs

Pectin is a large polysaccharide with a complex structure and function, which is composed of homogalacturonan (HG), rhamnogalacturonic acid and polygalacturonan (Gal) [26]. The biosynthesis of HG is catalyzed by *α*-1,4-D-galacturonosyltransferase (EC 2.4.1.43), designated by GAUT, and some GAUT-like (GATL) proteins [27]. In our experiment, four GAUT-like genes, named scaffold02462, scaffold00897, scaffold02056, and scaffold04120 were detected to be involved in the pectin biosynthetic process (GO: 0045489) (Table 4). Scaffold00897 and scaffold04120 showed higher expression levels in 50% calcified chestnuts, while scaffold02462 and scaffold02056 displayed slightly higher expression levels in the 0% calcified samples. The expression of scaffold04120 (EC 2.4.1.43) was up-regulated 12-fold in the 50% calcified samples compared with the fresh control, but the other three GAUT-like genes displayed down-regulated expression at 50% calcification. Pectin degradation enzymes mainly include two types: esterase and polymerase. Pectin methylesterase (PME, E.C 3.1.1.11) is an enzyme that catalyzes the specific hydrolysis of the methyl ester bond at C-6 in Gal residues in the linear HG domain of pectin [28]. In this study, we detected three DEGs, scaffold11321, scaffold01256, and scaffold00325, involved in pectinesterase activity (GO: 0030599). Among them, scaffold11321 belongs to PME (EC 3.1.1.1) and presented a very high expression level at 0% calcification while a relatively low expression levels at 50% calcification. Polygalacturonase (PG, EC 3.2.1.15) is another important enzyme that breaks the *α*-1,4-linkage of polygalacturonan and participates in pectin degradation [29]. Here, scaffold02071 showed a high expression level at 0% and a low expression level at 50%, which was discovered to exhibit polygalacturonase activity (GO: 0004650). In general, all enzymes involved in pectin metabolism were down-regulated, except scaffold04120. Scaffold04120 was annotated as glycosyltransferase, CAZy family GT8 (EC 2.4.1.43); to some extent, it was assumed to be an important enzyme involved in chestnut calcification.

### 3.12. Validation of Transcriptome Data by qRT-PCR

Quantitative RT-PCR was performed and validated the RNA-seq derived expression. Sixteen DEGs were randomly selected for qRT-PCR analysis. The primers used in the amplification of each candidate are listed in Appendix A. The results showed that the expression patterns of the selected genes coincided with the RNA-seq data. The expression patterns of some genes at 0% and 50% calcification are described in Figure 10.

## 4. Discussion

Calcification is an important factor that affects chestnut quality during storage and transportation [30]. A previous study indicated that continuous water loss accounts for the occurrence of calcification [12]. In our study, when fresh chestnuts were placed at a room temperature of 20–25 °C and a relative humidity of 50–60% for 47 d, they were 100% calcified and completely lost their edible value. The chestnut calcification index and moisture content showed a negative correlation (r = −0.961). This indicates that chestnut dehydration is one of the main reasons for calcification, which means that controlling the reduction of moisture during chestnut storage can effectively slow down the calcification process and promote chestnut preservation.

Abiotic stress has an effect on the normal growth of plants, and the cell wall is the first defensive barrier used to address external stress. Cellulose, pectin, and other components of the cell wall participate in the physiological response to stress, helping to control temperature, salinity, drought, and water [31,32]. In this study, we found that during the process of chestnut calcification, CWM accumulated continuously, so that the CWM of 100% calcified chestnut was 5.3-fold higher than that of fresh chestnut. The main components of the cell wall of chestnut include cellulose, hemicelluloses, lignin, and pectin [9]. The cellulose content increased and peaked at 50% calcification; however, the cellulose content decreased slightly at 100% calcification but had still increased when compared with that in fresh fruit. The content of lignin peaked at 25% calcification and was the lowest at 50% calcification. The WSP of the chestnut in the present study changed slowly during the calcification process, while the contents of CSP and ISP increased after reaching the lowest level at 50% calcification, indicated that the CSP and ISP contents were proposed to be associated with the calcification occurrence of chestnut.

In our study, 0% and 50% calcified chestnut samples were collected and used for comparative transcriptome analysis. A total of 1801 DEGs, which included 805 up-regulated and 996 down-regulated genes, were identified. The KEGG pathways related to chestnut calcification included purine metabolism, RNA degradation, the mRNA surveillance pathway, starch and sucrose metabolism, arginine and proline metabolism, and fatty acid metabolism, and were found. These results provided insight into the molecular mechanisms involved in chestnut calcification.

Because RNA could not be extracted from the 100% calcified chestnuts, we hypothesized that RNA degradation occurred during chestnut calcification. At 50% calcification, the purine metabolism (KEGG pathway: ko00230), RNA degradation (KEGG pathway: ko03018), and mRNA surveillance pathways (KEGG pathway: ko03015) were significant enriched, which has been reported to be closely related to RNA degradation [33].

The cell wall serves as the outermost barrier of plant cells and plays a significant role in transferring sense stress signals into the cell which can regulate cell processes [34]. In response to stress, cell wall metabolism-related genes play important roles in cell wall structure, metabolism, and signal transduction [35]. The genes associated with cell wall metabolism and chestnut calcification are closely related. In the present study, we detected six genes related to the cellulose biosynthetic process, including a SUS-related gene scaffold0125, which was highly expressed in calcified chestnut. Many studies have reported that cellulose biosynthesis can change water deficit through a decrease in the cellulose content [36,37]. During fruit storage, synthesized lignin is deposited in the cell wall, leading to a tighter cell wall structure, which maintains the integrity of the cell wall and finally causes an increase in fruit firmness. The decrease in the cellulose content in the later stage of calcification may be a result of damage to the cell membrane. Because cellulose synthase is located on the cell membrane, calcification causes the destruction of the cell membrane, so that the activity of cellulose synthase is decreased, resulting in greater cellulose degradation than synthesis [38]. In the present study, the content of cellulose increased rapidly and peaked at 50% calcification, followed by a gradual decreased, which suggested that cell membrane damage had been initiated. However, the lignin content reached the lowest level at 50% calcification and started to increase with continuing calcification. The secondary cell wall can be strengthened by the incorporation of lignin. Lignification is a complex process that includes several enzymes and phenolic substrates, which can occur prematurely to avoid cell wall damage when plants are exposed to a long period of water stress deficit [39,40].

The presence of rich pectin polysaccharides prevents water loss during desiccation. In parallel, pectin-degrading genes, including PG, can be down-regulated by water stress, resulting in improved cell wall integrity and cell expansion [41]. Three GAUT-like genes involved in HG biosynthesis were found to participate in pectin biosynthesis, while three genes and a PG were found to participate in pectin degradation.

Polygalacturonase is an important enzyme in the fruit ripening period that degrades polygalacturonic acid into oligogalacturonic and galacturonic acids in the cell wall of the fruit, disintegrating the cell wall structure and causing the fruit to soften [42,43]. The PG activity increased during fruit ripening [44], and PG was inactive in the early ripening stage of peach fruit. With a decrease in peach fruit hardness, the PG activity increased rapidly. When the peach had completely ripened and softened, the PG was most active. The change in PG activity was negatively correlated with the change in fruit hardness [45]. In the present study, the PG enzyme-related gene was down-regulated, and PG enzyme activity may also show a downward trend during chestnut seed calcification, which is opposite to the phenomenon of increased PG activity observed during the ripening and softening of other fruit. The down-regulated expression of PG enzyme genes may potentially result in the continuous hardening of Chinese chestnut during storage.

Fatty acids and their compositions can maintain the fluidity and integrity of the plant cell membrane under water deficit [46]. An increase in the fatty acid content can help to positively maintain the stability of cell membranes. The DEGs that participate in fatty acid biosynthesis (KEGG pathway: ko00061), arachidonic acid metabolism (KEGG pathway: ko00590), and α-linolenic acid metabolism (KEGG pathway: ko00592) were all up-regulated at 50% calcification during storage (Appendix A). The accumulation of fatty acids included arachidonic acid, and α-linolenic acid may decrease the rate of cell membrane damage during the process of chestnut calcification.

Abiotic stress has also been reported to influence amino acid metabolism, especially during the biosynthesis or degradation of some amino acids [47]. Chestnut calcification is a process of continuous water loss. Plants suffer abiotic stress, especially water deficit, resulting in the accumulation of a large amount of proline (Appendix A), which can regulate the osmotic pressure of cells, reduce the water potential, maintain the subcellular structure, and alleviate damage to plants caused by environmental stress. Arginine, a precursor of polyamine synthesis, is also an important nitrogen storage substance in plants. A large number of studies showed that when plants are subjected to abiotic stress such as drought or increased levels of metals or salt, large amounts of polyamines will accumulate in the plants; the increase in the polyamine content was positively correlated with the resistance of plants to abiotic stress [48,49,50]. Glutathione is an important water-soluble antioxidant substance. An increase in glutathione metabolism and biosynthesis-related enzyme activity can increase the ability of plants to defend themselves against environmental stress, including free radical scavenging, peroxide reduction, cell signaling, and repair pathway regulation. Glutathione reductase (GR, EC 1.6.4.2), glutathione peroxidase (GP, EC 1.11.1.9) (GP, EC 1.11.2), and glutathione S-transferase (GST, EC 2.5.1.8) are three important enzymes involved in glutathione metabolism [51]. Changes in the activities of these three enzymes are closely related to plant resistance to environmental stress. In this study, we found that scaffold29287 (GP, EC 1.11.1.12), a gene annotated as GP, was up-regulated, which implies that the activity of the GP enzyme was enhanced.

## 5. Conclusions

In summary, this study showed that the relationship between calcification in chestnut and the content of moisture was negative correlated (r = −0.961). The main reason for the formation of calcification in chestnut was water deficit. At the same time, the content of CWM in 100% calcified chestnut was 5.3-fold higher than that of fresh chestnut, an accumulation of large amounts of cell wall material includes cellulose, lignin, and pectin that can delay the calcification process. The KEGG pathways related to chestnut calcification included purine metabolism, RNA degradation, the mRNA surveillance pathway, starch and sucrose metabolism, arginine and proline metabolism, and fatty acid metabolism, which were detected by transcriptome analysis. Furthermore, two important genes, scaffold11300 and scaffold0412, were annotated as cellulose and pectin synthase genes, respectively, which may contribute to the increase in the total accumulation of cell wall material during chestnut calcification. The results provide a sequential transcriptomic and metabolic integrated analysis of chestnut calcification during postharvest storage.

## Figures and Tables

**Figure 1 foods-11-01136-f001:**
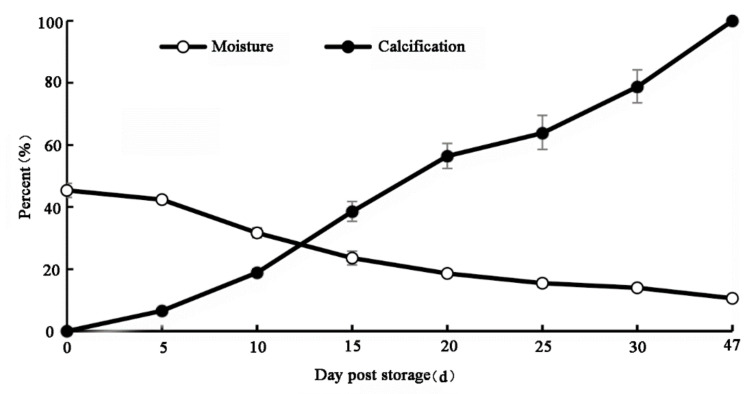
Relationship between the degree of chestnut calcification and the moisture content. Each experiment included three replicates. All of the data were expressed as means ± standard deviations.

**Figure 2 foods-11-01136-f002:**
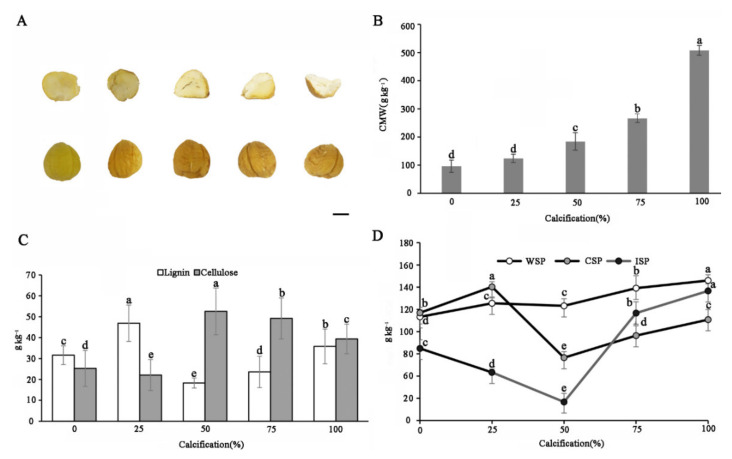
Changes in the cell wall content in different stages of chestnut calcification. (**A**) Degree of chestnut calcification, from left to right, 0%, 25%, 50%, 75%, and 100%. Scale bars, 1 cm. (**B**) Changes in cell wall material (CWM) in different stages of chestnut calcification. (**C**) Changes in lignin and cellulose contents in different stages of chestnut calcification. (**D**) Changes in the pectin content at different stages of chestnut calcification. All of the data were expressed as means ± standard deviations. Different letters (a–e) in the same composition indicate significant difference (*p* < 0.05, Duncan’s multiple range test).

**Figure 3 foods-11-01136-f003:**
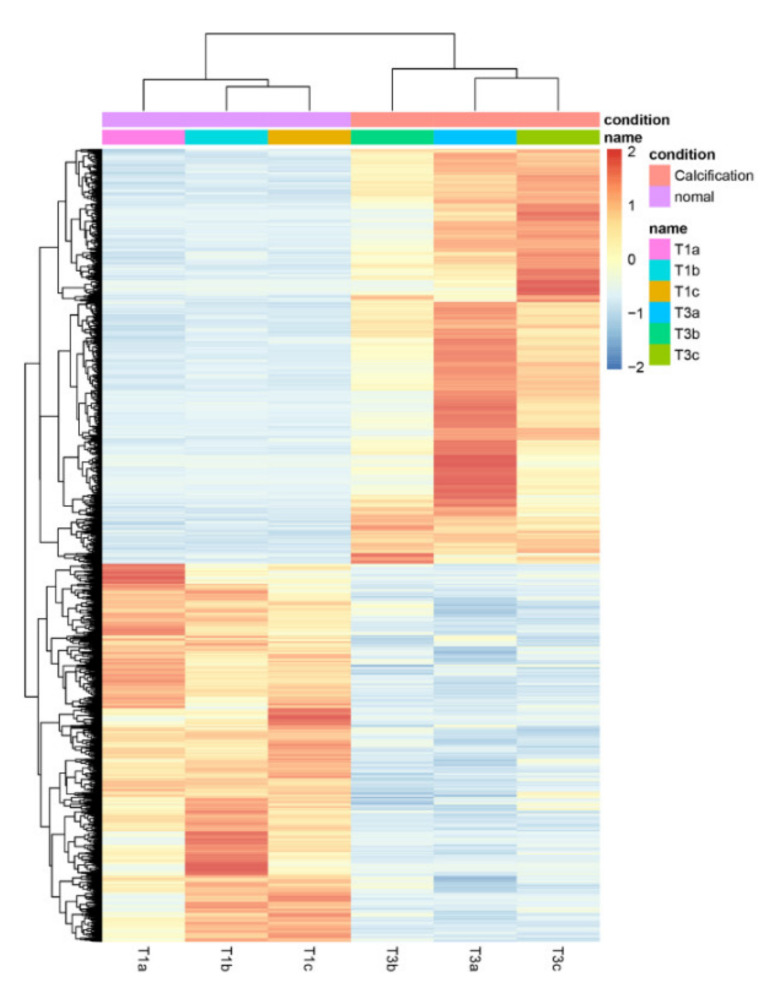
Heat maps of the differentially expressed genes (DEGs) in the 0% and 50% stages of chestnut calcification. The fragments per kb per million (FPKM) read values of unigenes were used for hierarchical cluster analysis. Expression levels are shown in different colors; the redder or bluer the color, the higher or lower the expression, respectively. The two groups, T1a, T1b, T1c and T3a, T3b, T3c, represent 0 % and 50% stages of chestnut calcification, respectively.

**Figure 4 foods-11-01136-f004:**
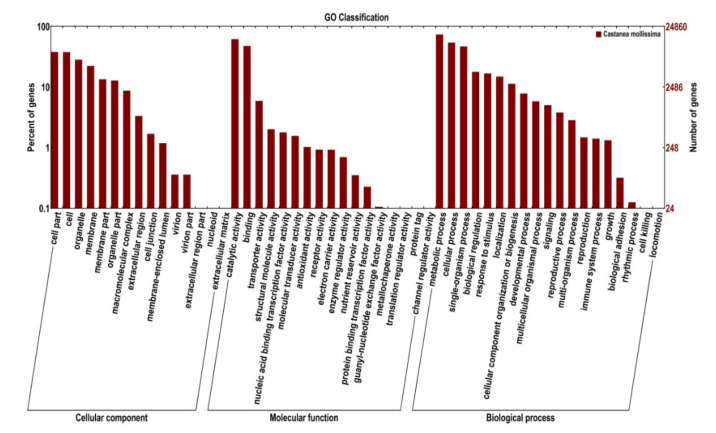
Gene ontology (GO) classification of differentially expressed genes between the 0% and 50% stages of chestnut calcification.

**Figure 5 foods-11-01136-f005:**
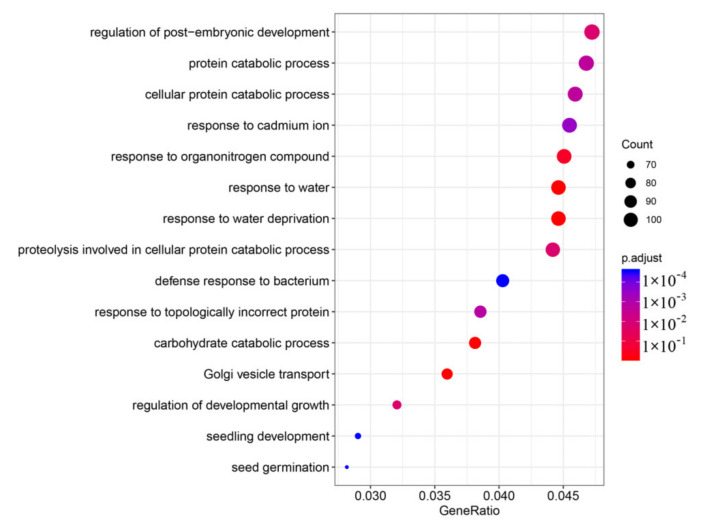
Gene ontology (GO) enrichment analysis of differentially expressed genes (DEGs).

**Figure 6 foods-11-01136-f006:**
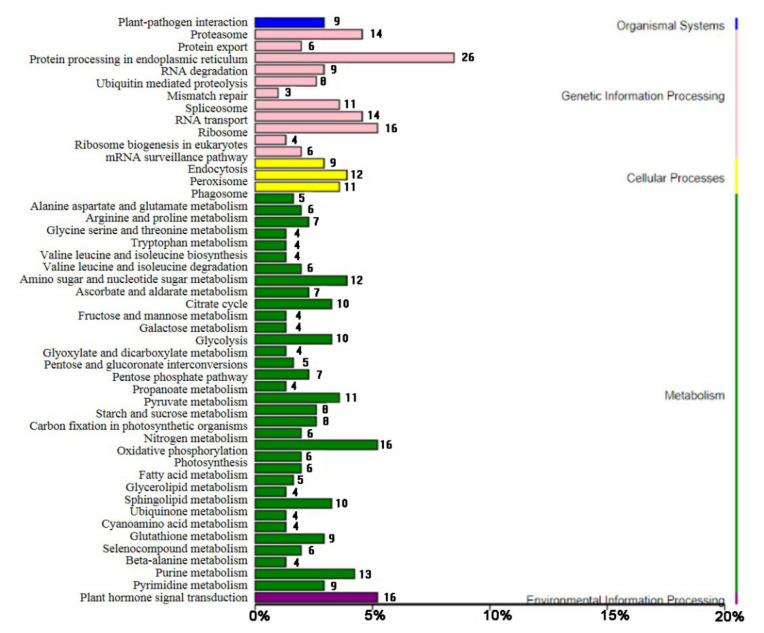
Kyoto Encyclopedia of Genes and Genomes (KEGG) enrichment analysis of differentially expressed genes (DEGs).

**Figure 7 foods-11-01136-f007:**
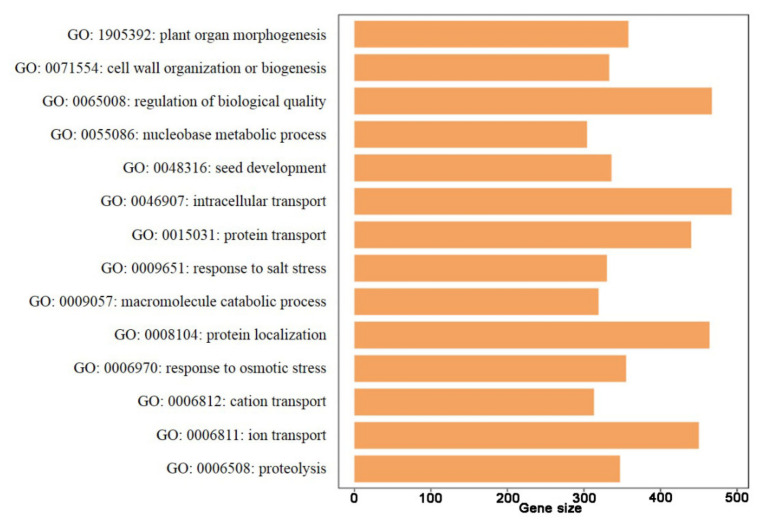
Gene set enrichment analysis (GSEA) enrichment analysis was performed base on GO annotations.

**Figure 8 foods-11-01136-f008:**
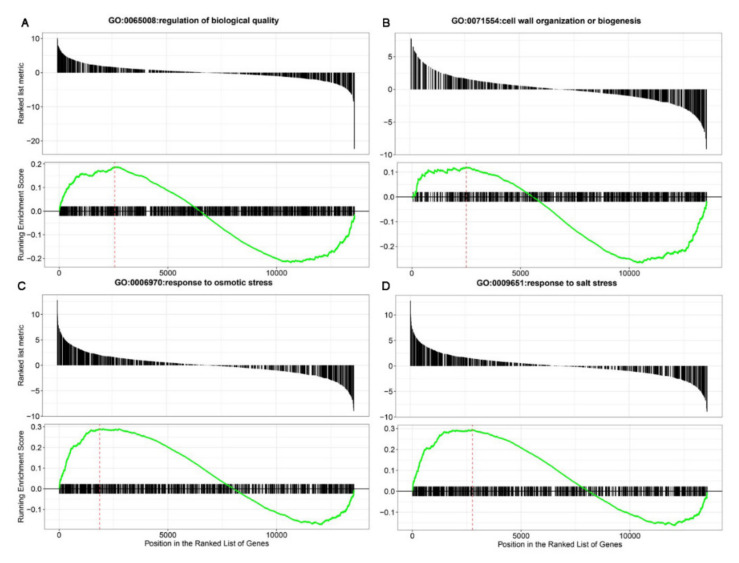
Gene set enrichment analysis (GSEA) analysis of differentially expressed genes (DEGs) expressed between the 0% and 50% calcification subtypes using the gene ontology (GO) datasets. (**A**) organization or biogenesis; (**B**) regulation of biological quality; (**C**) response to osmotic stress; (**D**) response to salt stress.

**Figure 9 foods-11-01136-f009:**
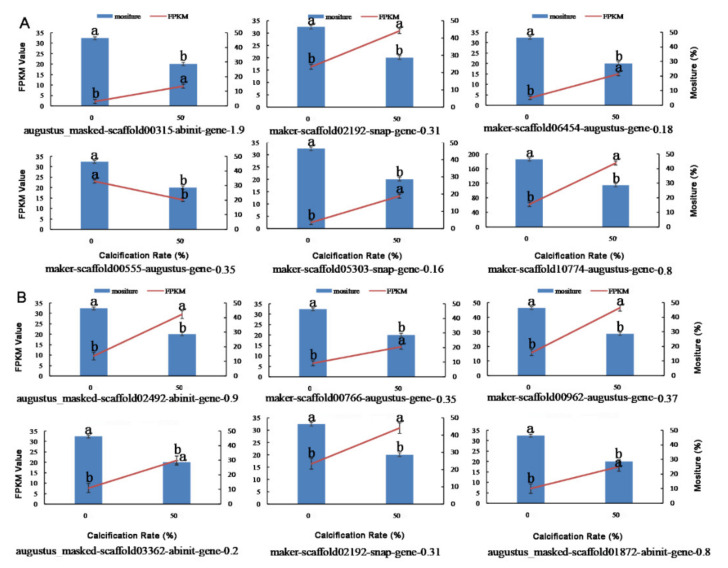
Relationship between the gene expression and the moisture content of chestnut calcification. (**A**) The differential expression genes (DEGs) in water transport (GO: 0006833) pathway; (**B**) the differential expression genes (DEGs) in response to water deprivation (GO: 0009414) pathway. Blue line: moisture content, red bars: transcriptomics RNA-seq analysis (FPKM). The letters (a,b) indicate significant differences (*p* < 0.05).

**Figure 10 foods-11-01136-f010:**
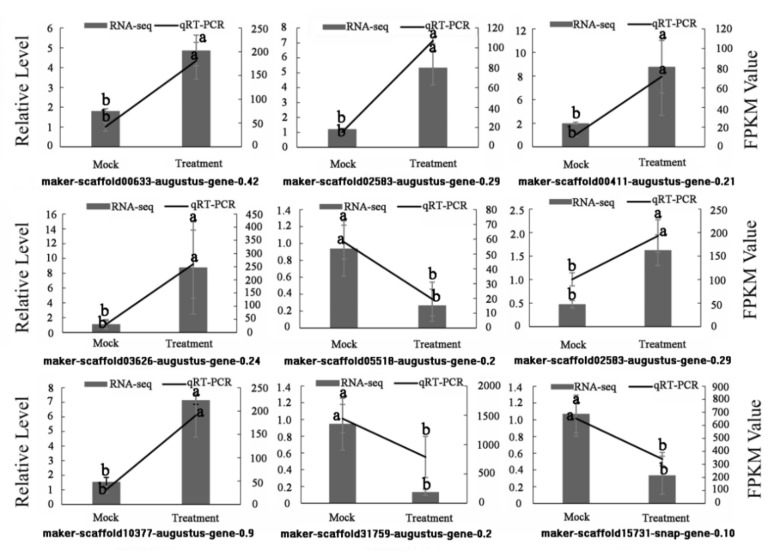
QRT-PCR validations of RNA-seq data. Relative expression was determined using three biological replicates. Black line: qRT-PCR expression, black bars: transcriptomics RNA-seq analysis (FPKM). The letters (a,b) indicate significant differences (*p* < 0.05).

**Table 1 foods-11-01136-t001:** Summary of the tested chestnut transcriptome sequencing dataset.

Samples	Clean Reads	Clean Data	GC content (%)	≥Q30
T1a	32,619,737 ^b^	8,154,934,250 ^b^	44.91	93.50
T1b	30,425,950 ^d^	7,606,487,500 ^d^	44.95	93.57
T1c	23,793,560 ^e^	5,948,390,000 ^e^	45.06	93.13
T3a	31,853,069 ^c^	7,963,267,250 ^c^	45.22	93.73
T3b	34,018,941 ^a^	8,504,735,250 ^a^	45.29	93.36
T3c	32,735,082 ^b^	8,183,770,500 ^b^	45.06	93.36

Notes: Q30 indicates the percentage of data/bases with a Phred quality score >30. A Phred quality score >30 means > 9.9% base call accuracy. The letters (a–e) in the same column indicate significant differences (*p* < 0.05).

**Table 2 foods-11-01136-t002:** Summary of the chestnut transcriptome reads mapped to the reference genes.

Samples	Total Reads	Mapped Reads	Unique Mapped	Multiple Mapped
T1a	65,239,474 ^b^	54,516,406 ^c^	53,664,614 ^c^	851,792 ^b^
T1b	60,851,900 ^d^	50,891,629 ^d^	49,885,627 ^d^	1,006,002 ^a^
T1c	47,587,120 ^e^	39,889,854 ^e^	39,123,856 ^e^	765,998 ^c^
T3a	63,706,138 ^c^	54,636,733 ^c^	53,975,213 ^c^	661,520 ^d^
T3b	68,037,882 ^a^	58,347,962 ^a^	57,572,054 ^a^	775,908 ^c^
T3c	65,470,164 ^b^	56,715,906 ^b^	56,031,375 ^b^	684,531 ^d^

The letters (a–e) in the same column indicate significant differences (*p* < 0.05).

**Table 3 foods-11-01136-t003:** Genes involved in cellulose and lignin metabolism.

Gene	FPKM (0%)	FPKM (50%)	FDR	Log2FC	Regulated
Cellulose biosynthetic process					
Scaffold01281-gene-0.39	7.13	1.59	0.0034	−1.84	Down
Scaffold01202-gene-0.30	112.48	42.79	0.00012	−2.00	Down
Scaffold03287-gene-0.16	2.05	0.43	0.00100	−4.46	Down
Scaffold00980-gene-0.41	0.08	0.03	0.00681	−1.51	Down
Scaffold02563-gene-0.4	33.76	10.17	0.00549	−1.95	Down
Scaffold11300-gene-0.9	4.83	16.05	0.00676	1.73	Up
Sucrose synthase activity					
Scaffold0125-gene-0.31	78.45	30.52	0.00358	−1.44	Down
Lignin catabolic process					
Scaffold34956-gene-0.3	8.24	1.04	0.00128	−3.09	Down
Scaffold08019-gene-0.16	0.71	0.05	0.00021	−4.35	Down

**Table 4 foods-11-01136-t004:** Genes involved in pectin metabolism.

Gene	FPKM (0%)	FPKM (50%)	FDR	Log2FC	Regulated
Pectin biosynthetic process					
Scaffold02462-gene-0.13	1.15	0.13	1.34 × 10^−7^	−3.60	Down
Scaffold00897-gene-0.33	20.93	7.75	0.00201	−1.53	Down
Scaffold02056-gene-0.28	2.7	0.28	0.00554	−2.74	Down
Scaffold04120-gene-0.22	2.19	25.97	4.74 × 10^−6^	3.66	Up
Pectinesterase activity					
Scaffold11321-gene-0.6	33.66	0.22	2.92 × 10^−16^	−7.86	Down
Scaffold01256-gene-0.30	33.61	0.02	4.48 × 10^−9^	−4.36	Down
Scaffold00325-gene-0.6	0.25	0.03	0.00563	−4.85	Down
Polygalacturonase activity					
Scaffold02071-gene-0.36	6.75	1.15	0.00016	−2.43	Down

## Data Availability

Data is contained within the article or Appendix A.

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
