# Peer review of "Transcriptomics Integrated with Changes in Cell Wall Material of Chestnut (Castanea mollissima Blume) during Storage Provides a New Insight into the “Calcification” Process"

_foods, 2022, doi:10.3390/foods11081136_

Round 1
Reviewer 1 Report
The manuscript is interesting, but the authors should use a means technique comparison as tukey or Duncan´s in results of figure 2 (b, c and d)
Reviewer 1 Report
The manuscript is interesting, but the authors should use a means technique comparison as tukey or Duncan´s in results of figure 2 (b, c and d)
Author Response
Thanks for the suggestion. At the revised manuscript, we used a means technique comparison of Duncan´s test in results of figure 2 (b, c and d). Please see line 240.
Reviewer 2 Report
The manuscript entitled ‘Transcriptomics integrated with changes in cell wall material of 2 chestnut (Castanea mollissima Blume) during storage provides a 3 new insight into the “calcification” process’ focused on the calcification phenomenon in chestnut seed. The relationship between the degree of calcification and several physicochemical parameters, including moisture, cell wall material contents, contents of cellulose, lignin and pectin were measured and studied. Transcriptome analysis was conducted to provide insights into the metabolisms related to the calcification phenomenon. In general, the manuscript reports very interesting findings, the data reported supports the conclusion. However, the manuscript needs be further improved to meet the publication criteria of the journal. The authors should consider the following suggestions:
- The language of the written English in the manuscript needs significant improvement. The tense and the grammar in the entire manuscript need to be checked and revised carefully. The authors should seriously consider to have a professional English editing service to polish the language to improve the readability and meet the criteria of the journal.
- Supplementary files are not found in the submitted document.
- Line 19: The phrase ‘response to’ should be written only one time.
- Line 19-20: The abbreviations should be spelled out the first time they appeared in the manuscript.
- Line 27-28: The logic of this sentence is not clear. The result does not provide a analysis, the analysis provide certain results. The authors should also include a sentence to better illustrate the significance and values of this study.
- Line 32: All the Latin names should be written using italic font, and they should be consistent throughout the manuscript.
- Line 81: Could the authors be more specific in explaining 'uniform'? What did the author mean by 'spread out'? In single layer?
- Line 83: Aren't the chestnuts already shelled before storage? What kind of samples were used for storage? Chest in-shell or only the kernel?
- Line 92-98: The description is not clear, could the authors illustrate the moisture content measurement procedure with a picture? Please also add an equation to show how the moisture content was calculated.
- Line 103: does gm means gram? if so, please use the SI unit g
- Line 103-104: Did the authors mean the degree of calcification was controlled exactly at 25%, 50%, 75% and 100%? How was it controlled? The authors claimed that they collected samples every 3 days, would that guarantee and control the degree of calcification at these levels? Please illustrate clearly.
- Section 2.3.2: Please add details of the measurement methods of lignin and cellulose, just like the authors did for section 2.3.3. Not all readers know exactly the methods cited.
- Section 2.4.5: what kind of statistical analysis were performed to determine the significance of difference in this study? Analysis of variance? Correlation analysis? Mean comparison test? Please add necessary information.
- Line 169: Please add the company name, town, state and country information of Excel software.
- Line 174: percentage is a relative and vague term, it is very confusing to use percentage for both the moisture content and relative change, the authors should include the equation used to calculate the moisture content, and change the unit of moisture to g water/g dry mass.
- Section 3.2: The title is not clear. It should be clearly stated as change in the cell wall material content to avoid confusion, since the authors introduced only the overall contents of cell wall material here, the change in chemical compositions in cell wall were discussed in a separate session.
- Section 3.3: It is suggested that the authors show the correlation coefficient between the lignin content and cellulose content, as the authors claimed that they have done correlation analysis.
- Line 223: key time or calcification degree? Will the findings be valid outside the tested range of conditions than in this study?
- Figure 2: the statistical analysis results in Figure 2 are not clear. For example, in Figure 2C, the same ** sign was observed in the '25% lignin' and '50% celllulose' and '50% lignin', which seem significantly different, and were confusing. The authors need to state clearly how the statistical analysis were conducted, and how what were compared in the mean comparison tests.
- Line 232: P should be in lower case letter
- Table 1 and 2: can the authors show statistical analysis results in Table 1 and 2?
- Figure 9 and 10: Statistical analysis results are not found, please add.
- Section 5 Conclusion: The authors should consider adding some quantitative results related to the major findings from this study in the conclusion.
Author Response
Response to Reviewer 2 Comments
Reviewer 2
The manuscript entitled ‘Transcriptomics integrated with changes in cell wall material of 2 chestnut (Castanea mollissima Blume) during storage provides a 3 new insight into the “calcification” process’ focused on the calcification phenomenon in chestnut seed. The relationship between the degree of calcification and several physicochemical parameters, including moisture, cell wall material contents, contents of cellulose, lignin and pectin were measured and studied. Transcriptome analysis was conducted to provide insights into the metabolisms related to the calcification phenomenon. In general, the manuscript reports very interesting findings, the data reported supports the conclusion. However, the manuscript needs be further improved to meet the publication criteria of the journal. The authors should consider the following suggestions:
Point 1: The language of the written English in the manuscript needs significant improvement. The tense and the grammar in the entire manuscript need to be checked and revised carefully. The authors should seriously consider to have a professional English editing service to polish the language to improve the readability and meet the criteria of the journal.
Response 1: Based on your comments, We improved the language and gramma of the revised manuscript by LetPub (http://www.letpub.com.cn/), please see the attached files below, hopefully, the present version of the manuscript might meet the criteria of the journal.
Point 2: Supplementary files are not found in the submitted document.
Response 2: Thank you for your comments. We have re-submitted the document of supplementary files in the submission system.
Point 3: Line 19: The phrase ‘response to’ should be written only one time.
Response 3: Thank you for your comments. We have revised it. Please see line 19.
Point 4: Line 19-20: The abbreviations should be spelled out the first time they appeared in the manuscript.
Response 4: Thank you for your comments. We have added the abbreviations of some words, e.g. gene ontology (GO), gene set enrichment analysis (GSEA), Kyoto Encyclopedia of Genes and Genomes (KEGG).
Point 5: Line 27-28: The logic of this sentence is not clear. The result does not provide a analysis, the analysis provides certain results. The authors should also include a sentence to better illustrate the significance and values of this study.
Response 5: Thank you for your suggestions. We have replaced “The results provided new insights into chestnut calcification process and laid a foundation for further chestnut preservation” with “The results provide a sequential transcriptomic and metabolic integrated analysis of chestnut storage” to better illustrate the significance and values of this study following the suggestions of the respected reviewer.
Point 6: Line 32: All the Latin names should be written using italic font, and they should be consistent throughout the manuscript.
Response 6: Thank you for your comments. the suggestions have been incorporated into the revised manuscript.
Point 7: Line 81: Could the authors be more specific in explaining 'uniform'? What did the author mean by 'spread out'? In single layer?
Response 7: Thank you for your comments. To ensure the repeatability of the experiments, we used the chestnut materials with same size,which were described as ‘uniform ’, and the seeds of chestnuts were spread out in single layer to make the materials in same experiment conditions (temperature and humidity).
Point 8: Line 83: Aren't the chestnuts already shelled before storage? What kind of samples were used for storage? Chest in-shell or only the kernel?
Response 8: We stored chestnuts with the shell, and removed the shell of seeds when checked for chestnut calcification, the details of materials sampling referenced to Xiao et al., 2021 (Scientia Horticulturae 289 (2021) 110473)
Point 9: Line 92-98: The description is not clear, could the authors illustrate the moisture content measurement procedure with a picture? Please also add an equation to show how the moisture content was calculated.
Response 9: Thank you for your comments. The moisture content measurement procedure of moisture content was reference to Rao et al., 2006, p43 (ISBN:978-92-9043-740-6). Tanaka et al., 2007 (doi.org/10.11449/sasj1971.38.127) and Babiker et al., 2010 (doi.org/10.5897/AJPS.9000095).The equation were added in the revised manuscript to show the calculating of moisture content.
Point 10: Line 103: does gm means gram? if so, please use the SI unit g
Response 10: Thank for your suggestions, we pologize for the wrong written ‘gm’, which has been instead by ‘g’.
Point 11: Line 103-104: Did the authors mean the degree of calcification was controlled exactly at 25%, 50%, 75% and 100%? How was it controlled? The authors claimed that they collected samples every 3 days, would that guarantee and control the degree of calcification at these levels? Please illustrate clearly.
Response 11: Thank you for your comments. To make the story of the present study more logical, we selected the seeds with 25%, 50%, 75% and 100% calcification for the physiological data measurement and RNA-sequencing, before the sampling, We did a pre-test where the samplings were every 3 d and 5 d, the results shown a little variation of the calcification degree in 3d, however, the degree of calcification changed obviously in 5 d (Fig 1), so we collected samples every 5 d. we have thoroughly corrected the sample details in the revised manuscript to make it more consistent and logical.
Point 12: Section 2.3.2: Please add details of the measurement methods of lignin and cellulose, just like the authors did for section 2.3.3. Not all readers know exactly the methods cited.
Response 12: Thank you for your comments. The methods has have been added in the revised manuscript.
Point 13: Section 2.4.5: what kind of statistical analysis were performed to determine the significance of difference in this study? Analysis of variance? Correlation analysis? Mean comparison test? Please add necessary information.
Response 13: Thank you for your comments. One way analysis of variance (ANOVA) was employed to determine the statistical difference. Significant differences between means were identified using Duncan’s multiple range test (p < 0.05).
Point 14: Line 169: Please add the company name, town, state and country information of Excel software.
Response 14: Thank you for your comments. We have added the company name, town, state and country information of Excel software.
Point 15: Line 174: percentage is a relative and vague term, it is very confusing to use percentage for both the moisture content and relative change, the authors should include the equation used to calculate the moisture content, and change the unit of moisture to g water/g dry mass.
Response 15: Thank you for your comments. The equation has been added, the methods and unit of moisture were reference to Rao et al., 2006, p43 (ISBN:978-92-9043-740-6). Tanaka et al., 2007 (doi.org/10.11449/sasj1971.38.127) and Babiker et al., 2010 (doi.org/10.5897/AJPS.9000095).Where SMC (%) = wet weight – dry weight × 100/wet weight. In our study the moisture content of seeds was determined twice; the samples were weighting immediately on sampling (fresh seeds) and after drying.
Point 16: Section 3.2: The title is not clear. It should be clearly stated as change in the cell wall material content to avoid confusion, since the authors introduced only the overall contents of cell wall material here, the change in chemical compositions in cell wall were discussed in a separate session.
Response 16: Thank you for your comments. We have corrected it this section in the revised manuscript.
Point 17: Section 3.3: It is suggested that the authors show the correlation coefficient between the lignin content and cellulose content, as the authors claimed that they have done correlation analysis.
Response 17: Thank you for your comments. The correlation coefficient analysis between the lignin and cellulose content was added in revised manuscript.
Point 18: Line 223: key time or calcification degree? Will the findings be valid outside the tested range of conditions than in this study?
Response 18: Thank you for your comments. The key time of chestnut calcification degree was calculate based on the results of our study (biochemical parameters and gene expression levels of different calcificated seeds). We think the findings could be valid outside the tested range of conditions than in this study.
Point 19: Figure 2: the statistical analysis results in Figure 2 are not clear. For example, in Figure 2C, the same ** sign was observed in the '25% lignin' and '50% celllulose' and '50% lignin', which seem significantly different, and were confusing. The authors need to state clearly how the statistical analysis were conducted, and how what were compared in the mean comparison tests.
Response 19: Thank you for your comments. We use Duncan’s multiple range test (p < 0.05) in our statistical analysis. Different letters (a–e) in the same composition indicate significant difference (p < 0.05). We have removed “**” sign and replaced with letter “a-e”. We compared 0, 25%, 50%, 75%, and 100% calcification of lignin and the same as celllulose. The result were shown in the revised manuscript.
Point 20: Line 232: P should be in lower case letter
Response 20: Thank you for your comments, we corrected “P” to “p”.
Point 21: Table 1 and 2: can the authors show statistical analysis results in Table 1 and 2?
Response 21: Thank you for your comments. We have showed statistical analysis results in revised Table 1 and 2.
Point 22: Figure 9 and 10: Statistical analysis results are not found, please add.
Response 22: Thank you for your comments. Statistical analysis results this has been added.
Point 23: Section 5 Conclusion: The authors should consider adding some quantitative results related to the major findings from this study in the conclusion.
Response 23: Thank you for your kind suggestions. We have added some quantitative results to the conclusion part of the manuscript.

Reviewer 3 Report
This study investigates the changes in the cell wall material of chestnut (Castanea mollissima Blume) during storage connected with the “calcification” process. Additionally, the transcriptome analysis was performed and potential pathways related to chestnut „calcification” were indicated. The topic of this study is very interesting and follows the current trends in foodomics. The introduction provides sufficient background and includes all relevant references. Experiments are generally well designed and properly described. Modes of results presentation can be improved - figures 3, 4, 5, 6, 8, 9 and 10 should be enlarged because they are difficult to read. Discussion is supported by results. Conclusions summarizing the most important findings. Furthermore, the references style should be revised and unified.
Reviewer 3 Report
This study investigates the changes in the cell wall material of chestnut (Castanea mollissima Blume) during storage connected with the “calcification” process. Additionally, the transcriptome analysis was performed and potential pathways related to chestnut „calcification” were indicated. The topic of this study is very interesting and follows the current trends in foodomics. The introduction provides sufficient background and includes all relevant references. Experiments are generally well designed and properly described. Modes of results presentation can be improved - figures 3, 4, 5, 6, 8, 9 and 10 should be enlarged because they are difficult to read. Discussion is supported by results. Conclusions summarizing the most important findings. Furthermore, the references style should be revised and unified.
Author Response
Thank you for your carefully and insightful review. We have improved and enlarged figures 3, 4, 5, 6, 8, 9 and 10, the references also has been corrected in the revised manuscript.
Round 2
Reviewer 2 Report
The authors have adequately addressed my questions. The manuscript is acceptable for publication.